# Elicitation of *Bacillus cereus*-Amazcala (*B.c*-A) with SiO_2_ Nanoparticles Improves Its Role as a Plant Growth-Promoting Bacteria (PGPB) in Chili Pepper Plants

**DOI:** 10.3390/plants11243445

**Published:** 2022-12-09

**Authors:** Noelia I. Ferrusquía-Jiménez, Beatriz González-Arias, Alicia Rosales, Karen Esquivel, Eleazar M. Escamilla-Silva, Adrian E. Ortega-Torres, Ramón G. Guevara-González

**Affiliations:** 1Biosystems Engineering Group, Center of Applied Research in Biosystems (CARB-CIAB), School of Engineering, Autonomous University of Querétaro, Campus Amazcala, Carr, Amazcala-Chichimequillas Km 1.0, El Marqués 76265, Mexico; 2Graduate and Research Division, Engineering Faculty, Autonomous University of Queretaro, Cerro de las Campanas, Santiago de Queretaro 76010, Mexico; 3Chemistry Faculty, Autonomous University of Queretaro, Cerro de las Campanas, Santiago de Queretaro 76010, Mexico; 4Department of Chemical Engineering, National Technological Institute of Mexico-Celaya, Ave, Tecnologico y A. Garcia-Cubas, S/N, Col. Fovissste, Celaya 38010, Mexico

**Keywords:** *Bacillus* spp., eustress, PGPB, SiO_2_-NPs, nanotechnology, chili pepper

## Abstract

Agriculture needs to decrease the use of agrochemicals due to their high toxicity and adopt new strategies to achieve sustainable food production. Therefore, nanoparticles (NPs) and plant growth-promoting bacteria (PGPB) have been proposed as viable strategies to obtain better crop yields with less environmental impact. Here, we describe the effect of silica nanoparticles (SiO_2_-NPs) on survival, antioxidant enzymatic activity, phosphate solubilization capacity, and gibberellin production of *Bacillus cereus*-Amazcala (*B.c*-A). Moreover, the effect of the co-application of SiO_2_-NPs and *B.c*-A on seed germination, physiological characteristics, and antioxidant enzymatic activity of chili pepper plants was investigated under greenhouse conditions. The results indicated that SiO_2_-NPs at 100 ppm enhanced the role of *B.c*-A as PGPB by increasing its phosphate solubilization capacity and the production of GA7. Moreover, *B.c*-A catalase (CAT) and superoxide dismutase (SOD) activities were increased with SiO_2_-NPs 100 ppm treatment, indicating that SiO_2_-NPs act as a eustressor, inducing defense-related responses. The co-application of SiO_2_-NPs 100 ppm and *B.c*-A improved chili pepper growth. There was an increase in seed germination percentage, plant height, number of leaves, and number and yield of fruits. There was also an increase in CAT and PAL activities in chili pepper plants, indicating that bacteria–NP treatment induces plant immunity.

## 1. Introduction

Today, increasing the production of quality food is a global priority [1] due to the constant population growth and the severe problems caused by climate change, which has made agriculture unsustainable by the greater use of resources and energy, threatening food security worldwide [2,3]. On the one hand, to meet the food demand, strategies such as using chemical fertilizers, pesticides, and insecticides have been preferred by farmers due to their rapid action [1]. However, excessive use of chemical fertilizers causes problems such as eutrophication, root weakening, and soil acidification [4]. On the other hand, pesticide and insecticide residues have been identified in a large number of food products, so there is a big concern in society about the use of these chemicals due to their high toxicity [5]. Therefore, the research of sustainable agricultural strategies, such as plant growth-promoting bacteria (PGPB) and plant growth-promoting factors (PGPF), which allow farmers to obtain better yields and while having less environmental impact, is of great interest [5,6]. These alternatives propose a more efficient use of fertilizers, a decrease in the incidence of pests and diseases, and improvements in the nutraceutical quality of food [1].

PGPB are a diverse group of bacteria that promote plant growth through different mechanisms, such as the production of plant hormones, increased nutrient viability, induction of plant resistance, induction of biosynthesis of osmoprotectant molecules, and the production of hydrolytic enzymes; all these mechanisms help the plant to cope with different forms of biotic stress, such as the attack of pathogenic organisms, and abiotic stresses, such as drought and salinity [7,8]. Among the genera that contain species identified as PGPB are *Pseudomonas* spp., *Bacillus* spp., *Xanthomonas* spp., *Streptomyces* spp., *Agrobacterium* spp., *Arthrobacter* spp., *Enterobacter* spp., and *Rhizobium* spp. [9]. Within these, the species of the genus *Bacillus* have become highly relevant in agriculture. For example, *Bacillus cereus* YL6 improved yield in soybean (*Glycine max* L.), wheat (*Triticum vulgare* L.), and cabbage (*Brassica rapa* subsp. *Pekinensis*); this is due to its ability to solubilize organic and inorganic phosphorus and the production of phytohormones such as indole-3-acetic (IAA) and gibberellin (GA) [10]. *B. amyloliquefaciens SN13* promoted growth and induced tolerance to salinity and stress in rice [11]. *Bacillus cereus*-Amazcala (*B.c*-A) is a recently discovered strain by our laboratory that has demonstrated the ability to solubilize inorganic phosphate and produce gibberellic acid (GA3); in addition, it promoted an increase in biomass and a decrease in disease severity caused by *Clavibacter michiganensis* on tomato [12].

PGPF, also defined as eustressors, are stress factors of physical, biological (elicitors), and chemical origin that, under a specific dose, favor the development and yield of crops as well as the induction of the plant immune response, which includes responses such as the activation of antioxidant enzymes such as catalase (CAT), superoxide dismutase (SOD), and phenylalanine ammonia lyase (PAL) [13]. Furthermore, nanotechnology has offered important alternatives regarding using silica nanoparticles (SiO_2_-NPs) as eustressors [6]. Due to their small size (5–20 nm) and other characteristics, such as shape, surface charge, and texture, SiO_2_-NPs can easily penetrate the cell wall and distribute into tissues. It was observed that plants uptake SiO_2_-NPs through their roots and leaves by diffusion to later be polymerized or translocated into other tissues, producing different physiological and cellular functions [14,15]. SiO_2_-NP accumulation provides mechanical strength to the cell wall, functioning as a physical barrier against pathogens and abiotic stress factors [16]. Moreover, SiO_2_-NPs function as modulators of gene expression, controlling different signaling pathways that involve growth regulators such as salicylic acid (SA), jasmonic acid (JA), and ethylene. Therefore, they may promote the induction of different biochemical and physiological processes, enhancing plant growth and yield. Different studies have shown that SiO_2_-NPs, when applied to plants, increase lignification, biomass, height, flowering, nutrient acquisition, water absorption, germination rate, and chlorophyll and antioxidant enzyme content, improving photosynthetic efficiency, global yield, and disease management; moreover, they are viable for use in agriculture since they are nontoxic in nature [16]. For example, SiO_2_-NP application has shown improvements in germination and shoot and root growth in barley and maize [17]. In addition, applying SiO_2_-NPs at a dose of 8 g/L showed an increase in the germination percentage in tomato plants [18]. Similarly, studies have shown that SiO_2_-NPs effectively regulate the stress caused by UV-B radiation in wheat (*Triticum aestivum* L.) [19]. However, the interaction of SiO_2_-NPs with plant cells continues to be studied, due to their high complexity and variability in terms of doses and forms of application.

Beneficial effects of SiO_2_-NPs have also been reported in soil microbial communities. For example, it was observed that the use of silica nanoparticles promoted the increase in the total biomass of the soil microbial communities and, therefore, improved corn (*Zea mays* L.) growth [20]. Moreover, applying SiO_2_-NPs increased the population of the bacterial communities *Rhodobacteraceae* and *Paenibacillus* and *Chaetomium* fungal genera involved in soil nutrient recycling in pakchoi (*Brassica chinensis* L.) rhizosphere [21]. For this reason, it becomes relevant to evaluate the effect that SiO_2_-NPs may have on microorganisms such as PGPB.

Worldwide, Mexico is one of the primary producers of chili pepper (*Capsicum* spp.). Around 150,000 hectares are planted with chili pepper, generating a production of 3.2 million tons per year, placing the country as the leading exporter of chili pepper on an international scale and the second largest producer in the world [22]. About 75% of Mexican chili pepper producers use a large number of chemical pesticides to combat pests and diseases [23]. Thus, new natural and nontoxic strategies that allow less use of pesticides and sustainably increase crop yields are necessary.

Therefore, this study aimed to evaluate the in vitro effect of SiO_2_ nanoparticles on the role of *Bacillus cereus*-Amazcala (*B.c*-A) as PGPB as well as the application effect of *B.c*-A, in combination with SiO_2_-NPs on the growth and development of chili pepper. This study represents a novel alternative where the combination of *B.c*-A and SiO_2_-NPs could promote the development and growth of the chili crop, as well as a higher fruit yield. In addition, it was possible to observe that SiO_2_-NPs promote a better activity of *B.c*-A as PGPB, suggesting a role as a eustressor of soil microorganisms.

## 2. Results

### 2.1. Physicochemical Characterization of SiO_2_-NPs

TEM analysis was carried out to explore the morphology of the SiO_2_-NPs prepared by the sol–gel method. The micrograph shown in Figure 1A has been taken up to 200 nm index. The TEM image shows a cluster with a range of size of 200 µm conformed with small nanoparticles with an average diameter of ~8.9 nm showing amorphous morphology. This morphology and size are due to the increase in OH group concentration to reduce the rate of hydrolysis and condensation [24].

XRD analysis was carried out to determine the crystallinity of the SiO_2_-NPs, and the diffractogram of the nanoparticles is shown in Figure 1B. The XRD pattern shows the high-intensity peak of silicon dioxide at 24°, characteristic of the amorphous SiO_2_ phase [25,26]. Raman spectroscopy was carried out to complement the crystallinity analysis, and the recorded spectra of the SiO_2_-NPs are shown in Figure 1C. The vibrational modes corresponding to the amorphous phase of SiO_2_ are located at 493 and 605 cm^−1^, which are attributed to the three- and four-membered chains of silicon and oxygen. The band center around ~446 cm^−1^ is the characteristic band of glass. The band between 400 and 700 cm^−1^ corresponds to the inter-tetrahedral Si-O-Si bonding. The bands near 1000 and 1200 are attributed to the symmetrical stretching of silicon and oxygen. The bands near 1000 cm^−1^ and 1200 cm^−1^ are attributed to the symmetrical stretching of silicon and oxygen in the silicate tetrahedral [27,28].

### 2.2. In Vitro Effect of SiO_2_-NPs on B.c-A Survival

The in vitro application of SiO_2_-NPs (1, 10, and 100 ppm) on *B.c*-A was carried out to observe if there was any toxicity response. Interestingly, SiO_2_-NPs at 100 ppm did not produce adverse effects on the survival of the bacteria after 24 h of exposure, finding that the population was statistically similar to the control. However, there were negative effects when 1 and 10 ppm of SiO_2_-NPs were used, finding a statistically lower population than the control after 24 h of nanoparticle exposure (Figure 2A).

### 2.3. In Vitro Effect of SiO_2_-NPs on B.c-A CAT and SOD Activities

SiO_2_-NPs treatments showed an increase in the antioxidant enzyme activity of the bacteria. Both treatments, 10 and 100 ppm of SiO_2_-NPs, showed a significant increase in SOD and CAT activities compared to the control. The highest SOD and CAT activities were obtained with the treatment of 100 ppm of SiO_2_-NPs (Figure 2B,C).

### 2.4. In Vitro Effect of SiO_2_-NPs on B.c-A Phosphate Solubilizing Capacity

A significant increase in the phosphate solubilization halo was observed when exposing the bacteria to a concentration of 100 ppm of SiO_2_-NPs, indicating greater solubilization of phosphates (Figure 3A).

### 2.5. In Vitro Gibberellin Production by B.c-A in the Presence of SiO_2_-NPs

Regarding the production of gibberellins by *B.c*-A, the results showed that applying SiO_2_-NPs at 100 ppm stimulates the production of GA7. Moreover, the results showed that applying SiO_2_-NPs at the same concentration inhibited the production of GA1. Production of GA3 was not affected by SiO_2_-NP application (Figure 3B).

### 2.6. Effect of B.c-A in Combination with SiO_2_-NPs on Chili Pepper’s Growth, Development, and Antioxidant Activity

The germination results indicated that *B.c*-A + SiO_2_-NPs at 100 ppm treatment improved the germination percentage of chili seeds, obtaining 91% germination after 15 days, compared to the control, which had 66% germination at the same time (Figure 4A). 

Additionally, from day 7, early germination of seeds was observed using all treatments, with *B.c*-A + SiO_2_-NPs at 100 ppm showing higher amounts of early germination than the other evaluated treatments (Figure 4A).

To evaluate if the treatments induced any defense response in chili pepper plants, CAT and PAL activities were determined at 5 min after application and after one and two months of biweekly applications. The results showed that the CAT activity increased significantly throughout the application time with the *B.c*-A + SiO_2_-NPs at 100 ppm treatment (Figure 4B). Similarly, compared to the control, the SiO_2_-NPs at 100 ppm treatment also induced a significant increase in CAT activity after two months. In contrast, the PAL activity was increased significantly with SiO_2_-NPs at 100 ppm and *B.c*-A SiO_2_-NPs at 100 ppm treatments after two months (Figure 4C).

Regarding the evaluation of the physiological variables, it was observed that some treatments positively affected the development, growth, and quality of chili pepper. After eight weeks of biweekly applications of *B.c*-A + SiO_2_-NPs at 100 ppm, it was observed that the variables plant height, number of leaves, and fruits had a significant increase, while stem thickness was not significantly different from the control (Figure 5A–F). In addition, SiO_2_-NPs at 100 ppm treatment showed an increase in stem thickness and number of leaves compared to the control (Figure 5A,C). Regarding fruit yield, the application of SiO_2_-NPs at 100 ppm and *B.c*-A + SiO_2_-NPs at 100 ppm showed a significant increase compared to the control. Nevertheless, the highest yield was obtained with *B.c*-A + SiO_2_-NPs at 100 ppm (Figure 5E).

## 3. Discussion

The agricultural industry demands an urgent reduction of chemical fertilizers and pesticides based on environmental constrains caused by current agricultural practices. For this reason, alternatives such as the combined use of PGPB and nanoparticles are considered the technological and scientific future for the development of sustainable agriculture [29]. Data obtained from current research indicated that SiO_2_-NPs enhanced the activity of *B.c*-A as a PGPB and that co-application of *B.c*-A and SiO_2_-NPs results as a better strategy to improve growth, development, and yield in chili pepper crop.

Based on the survival results, it was observed that SiO_2_-NPs at 100 ppm are not toxic to *B.c*-A, while the application of lower concentrations decreased the viable cell count of *B.c*-A. This suggests that the effect of SiO_2_-NPs on the bacteria does not follow a hormetic pattern in which higher doses are more toxic than lower ones. This phenomenon is similar to that observed in plants since some NPs, including SiO_2_-NPs, can have negative impacts at specific doses and forms of application. Thus, hormetic studies in the evaluation of nanoparticles are essential [16]. Some studies have analyzed the effects of NPs on the growth and survival of PGPB. For example, the Au-NPs at 6.25 ppm application increased the growth of *Pseudomonas fluorescens*, *Bacillus subtilis*, and *Paenibacillus elgii* [30]. Moreover, studies showed that SiO_2_-NPs at 10 ppm increased soil populations of nitrogen fixers and phosphate solubilizers [31]. However, other studies have shown that NPs harm bacterial populations. For example, it was found that triangular Ag-NPs exhibited biocidal action in *E. coli* [32]. Thus, the toxic effects of NPs depend on the nature of the metal, shape, size, and concentration of NPs.

SOD and CAT activities were measured to evaluate the in vitro effect of SiO_2_-NPs on the induction of stress-related responses of *B.c*-A. Our results indicated that SiO_2_-NPs treatments increased *B.c*-A CAT and SOD activities, suggesting the activation of its antioxidant system, a usual defense response in organisms [13,33]. In bacteria, as in plants, the induction of the antioxidant system can occur due to several stress factors, such as temperature, chemicals, and nutrient availability. For example, an upregulation of genes encoding SOD y CAT in *B. subtillis* and *B. cereus* was observed after their interaction with mild and lethal concentrations of peracetic acid and heat shocks [33]. Therefore, similar to the effects observed in plants, our data suggest that SiO_2_-NPs at 100 ppm exert an eustressic effect on the bacteria by inducing their defense system without negatively affecting theirsurvival.

To obtain information regarding the effects of SiO_2_-NPs on *B.c*-A plant growth-promoting activity, the phosphate solubilizing activity and gibberellin production of *B.c*-A were evaluated. Our data showed that the SiO_2_-NPs at 100 ppm increased the phosphate solubilization activity and gibberellin (GA7) production of *B.c*-A. This set of responses suggests once again that SiO_2_-NPs act as a chemical eustress factor on *B.c*-A by improving its performance as PGPB. Phosphorus solubilization and gibberellin production are two beneficial features of PGPB widely studied [5,34]. Previous studies have analyzed the effects of NPs on the phosphate solubilization activity of PGPB. For example, it was observed that the application of SiO_2_-NPs at 100 ppm benefits the population of *Pseudomonas stutzeri* and *Mesorhizobium* spp., which in turn enhances nitrogen and phosphorus content in the soil and contributes to the increase in vegetative growth of land cress plants [35]. These findings are consistent with ours. Some explanations for this phenomenon may be that NPs act as a substrate or stimulate the growth and activity of nitrogen fixers and phosphorus-solubilizing bacteria [20,31,35]. However, more studies are needed to evaluate the biochemical and molecular processes of NP-bacteria interaction.

Regarding gibberellin production, to our knowledge, this is one of the first works to report that SiO_2_-NPs increase the production of gibberellins by PGPB. Nevertheless, in previous studies, it was observed that the production of IAA (indole-acetic acid) in *Pseudomonas chlororaphis* was enhanced with the application of CuO-NPs at 200 mg/L but inhibited by ZnO-NPs at 500 mg/L [36]. Thus, it can be expected that NPs have an effect on phytohormone production of PGPB. Other studies have focused on optimizing the metabolic processes of PGPB to enhance gibberellin production. For example, an optimization study using the response surface methodology (RSM) determined that a *B. cereus* isolate incubated for 5 days at 35 °C on pH 7, with an intake of de 0.1 g/L of ammonium chloride and 3 g/L of fructose, enhances GA production by up to 109.25 μg/mL [37].

As previously described, different studies support that NPs at specific doses improve the beneficial properties of PGPB, such as the synthesis of phytohormones, phosphate solubilization, nitrogen fixation, and growth [29,36,38]. Our work contributes to the understanding of this phenomenon, finding that the defense activity, gibberellin (GA7) production, and phosphate-solubilizing activity are elicited by the in vitro application of SiO_2_-NPs. This bacterial elicitation process can be understood as an advantage to face the challenges involved in using microbial inoculants in agriculture, which usually show inconsistencies in field application due to different factors that cause their destabilization. On the other hand, since the soil is an environment in which beneficial and pathogenic organisms develop, it can be suggested that nanoparticles can promote the activity of beneficial microorganisms and those of plant pathogens; however, different studies are needed to describe these interactions.

*B.c*-A elicitation with SiO_2_-NPs is consistent with the obtained results regarding the growth and development of the chili pepper crop. Our data indicate that the combined use of the bacteria and SiO_2_-NPs at 100 ppm increased some plant performance variables such as the germination percentage of chili pepper seeds by approximately 25%. Furthermore, this treatment increased the number of leaves, plant height, number of fruits, and total crop yield. Application of *B.c*-A or SiO_2_-NPs separately did not obtain the same results as their combined application, except for the SiO_2_-NPs at 100 ppm treatment, which promoted a significant increase in stem thickness. One possible explanation for this is that both the nanoparticles and the bacteria alone exert positive effects on the crop, but their combination also enhances the activity of the bacteria, so it is possible to find better results for growth, development, and crop yield. SiO_2_-NPs have a direct effect on plant cells since they are capable of modulating biochemical and physiological responses through the activation of genes related to the induction of growth regulators, the activation of the antioxidant system, and the generation of a mechanical barrier in the plant cell walls, which allows a better development, resistance, and vigor of the plants [16,39]. Moreover, similar responses have been observed with the combined use of PGPB and NPs. The co-application of TiO_2_-NPs and *Pseudomonas fluorescens* promoted the growth of *Trifolium repens* and increased its biomass, which in turn enhanced Cd uptake to promote soil phytoremediation [40]. In addition, inoculation of *Bacillus thuringiensis*, *Paenibacillus polymyxa*, *Alcaligenes faecalis*, and TiO_2_-NPs increased the biomass of wheat under adverse conditions [41]. Application of *Bacillus megaterium*, *Bacillus brevis*, *P. fluorescens*, *Azotobacter vinelandii*, and SiO_2_-NPs caused 100% of maize seed germination [31].

Furthermore, the application of *B.c*-A + SiO_2_-NPs 100 ppm induced an increase in CAT and PAL activities in chili pepper plants, which are defense-related enzymes that participate in oxidative stress signaling and the production of secondary metabolites (phenylpropanoids) to cope with biotic and abiotic stresses [42]. Interestingly, the activity of CAT was increased during the entire treatment application time, suggesting that the treatment causes a controlled elicitation of the crop and, therefore, likely constant protection against a possible attack of pathogens. Thus, co-application of PGPB and NPs is beneficial to help plants cope with biotic and abiotic stress. A recent study showed that the application of iron oxide nanoparticles and *Bacillus subtilis* S4 increased catalase (CAT), ascorbate peroxidase (APX), and superoxide dismutase (SOD), which mitigate arsenic stress in *Cucurbita moschata* L. [43]. Moreover, applying ZnO-NPs (20 and 40 mg L^−1^) and *Bacillus subtilis*, *Lactobacillus casei*, and *Bacillus pumilus* reduced genetic impairment caused by salinity stress in tomato seedlings [44]. It is essential to highlight that the application of nanoparticles or PGPB separately can promote certain benefits in crops; however, the accumulated information indicates that their combined application has an improved function. NPs can improve the beneficial attributes of PGPB such as nitrogen fixation, siderophore production, phosphorous solubilization, and phytohormone production [29] by stimulating their growth and activity; moreover, some authors report that NPs could function as platforms to stabilize bacteria attachment to roots [41], which in turn enhance plant growth and development. Furthermore, NPs alone have demonstrated their potential as plant growth-promoting factors, depending on their physicochemical characteristics such as nature (metallic, oxides, carbon-based, or composite), morphology, size, concentration, and crystallinity, by mentioning the main ones [16,45]. The double action (NPs–PGPB) allows this technology to acquire an important role in microbial formulation development. In addition, our results indicate that the co-application of *B.c*-A and NPs induces plant-immunity responses over time, which may result in a programmed elicitation scheme that promotes a better state of crop health.

Another relevant aspect is that many reports describe that nanoparticles can generate both positive and negative effects on plants and PGPB [29]. Therefore, it is essential to investigate the performance of NPs under a hormetic scheme based on testing different doses and exposure times for microbial formulation development with commercial purposes in agriculture. In our study, the synthesized SiO_2_-NPs used at 100 ppm did not cause toxicity in the bacteria or the chili pepper plants under the mentioned conditions. However, more studies will be necessary to verify if there is nanoparticle accumulation in plant tissues, as shown in other similar studies [6], as well as within *B.c*-A cells. Finally, our study provides an approach to the combined use of NPs and *B.c*-A, from which new research can be derived to understand the NP–*B.c*-A–plant interaction at a molecular level, as well as the new rationale design of PGPB-NPs commercial product formulations.

## 4. Materials and Methods

### 4.1. Biological Material 

A *Bacillus* strain, *B.c*-A (*B. cereus* “Amazcala”)**,** previously isolated and identified as beneficial by Solano-Alvarez et al. (2021) [12], was activated by its incubation in Luria Bertani liquid media (LB) during 24 h at 30° C under stirring (200 rpm). For plant bioassays, chili pepper (*Capsicum annuum* L.) seeds variety “Jalapeño Everman F1” (Harris Moran, seed company, Querétaro, México) was used.

### 4.2. SiO_2_ Nanoparticles Synthesis and Characterization

SiO_2_ nanoparticles (SiO_2_-NPs) were synthesized by the sol–gel method [24]. Briefly, tetraethyl orthosilicate (TEOS, 98%, Sigma Aldrich, St. Louis, MI, USA) was added dropwise to a 90% aqueous solution of absolute ethyl alcohol (EtOH, 98%, JT Baker) while magnetically stirring for 25 min. After this time, ammonium hydroxide (NH_4_OH, 28–30%, JT Baker) was used as an alkaline catalyzer by its dropwise addition to the solution until the solution pH was 10. The obtained gel was dried for 12 h at room temperature and calcinated at 100 °C for 30 min.

The SiO_2_-NPs were analyzed morphologically by transmission electron microscopy using a JOEL JEM-1010 (TEM) operating at 200 kV. The crystallinity of the SiO_2_-NPs was determined by X-ray diffraction (XRD) using a D8 diffractometer (Bruker, Billerica, MA, USA) equipped with a Cu anode to generate CuKα radiation (λ = 1.5406 Å) in an arrangement of 20° < 2θ < 80°. To complement the crystalline analysis, Raman spectroscopy was carried out with the LabRAM HR equipment (Horiba Scientific, Kyoto, Japan), which used an NdYGA laser (λ = 532 nm).

### 4.3. In Vitro Bioassays

*B.c*-A liquid cultures (1 × 10^7^ CFU/mL) established in LB media were incubated with SiO_2_-NPs at different concentrations, 1, 10, and 100 ppm, at 30 °C under stirring (200 rpm). Samples were taken at 24 h and processed for enzymatic activity determination and gibberellin production or cultured in LB solid media by triplicate to obtain CFU/mL, according to the most probable number method (MPN) [46].

To assess the phosphate solubilization capacity, the bacteria were inoculated with SiO_2_-NPs at 100 ppm on a Pikovskaya medium that contained calcium phosphate (CaPO_3_) as the only phosphate source. A volume of 5 μL of control (*B.c*-A liquid culture 1 × 10^7^ CFU/mL) and treated culture (*B.c*-A liquid culture 1 × 10^7^ CFU/mL + SiO_2_-NPs at 100 ppm) were inoculated in the agar and incubated during 10 days at 30° C. Formation of solubilization halos were measured with a vernier [47].

### 4.4. Quantification of Antioxidant Enzymatic Activities

Enzymatic extracts (EE) were previously obtained from recovered biomass (plant or bacteria). Bacterial cultures were centrifuged, and the obtained pellet was resuspended in phosphates buffer (pH 7.8). Resuspension was sonicated during 3 cycles of 20 s in an Elmasonic E 30 H sonicator. The tubes were centrifuged for 15 min. at 12,000 rpm, and the supernatant was collected as enzymatic extract (EE). EE was saved at 8 °C for enzymatic determination. Vegetal tissues were macerated with liquid nitrogen in a mortar until a fine powder was obtained. Then, 0.3 g of the powder was vortexed with 1 mL of phosphate buffer and centrifuged at 14,000 rpm for 15 min. The supernatant was collected and stored as EE [42].

For superoxide dismutase (SOD) activity, 0.05 mL of EE was added to a reaction mix containing 1.5 mL of phosphate buffer (50 mM, pH 7.8), 0.3 mL of nitro blue tetrazolium chloride (NBT) (0.75 mM), 0.3 mL of methionine (0.13 M), 0.3 mL of riboflavin, and 0.3 mL of EDTA (0.1 mM). The reaction tube was vortexed and exposed to uniform light of 12.5 lux for 15 min. Absorbance was read at 560 nm [48].

Catalase (CAT) activity was determined by mixing 2 mL of phosphate buffer (pH 7.8), 0.2 mL of 100 mM H_2_O_2_, and 0.1 mL of EE. Absorbance was measured at 240 nm each minute for 6 min, and the differences were used to calculate the enzymatic activity of CAT present in sample tissues [49].

Phenylalanine ammonia-lyase (PAL) activity was determined by mixing 0.2 mL of phosphate buffer with 0.1 mL of EE, and the mix was incubated for 60 min at 40 °C. Once the incubation time had elapsed, 50 μL of 1N HCl (JT Baker) solution was added (to stop the reaction) and let stand for 10 min. Then, the absorbance was measured at λ290 nm [50].

Spectrophotometric analyses were made in an SP-UV 1100-DLAB spectrophotometer. Total protein was determined by the Bradford method, and the enzymatic activity was expressed as Umg^−1^ of protein.

### 4.5. Quantification of Gibberellins

To evaluate gibberellin production, bacteria samples were taken and filtered in the cultured medium until pH 2.5, and gibberellins were extracted with ethyl acetate for further analysis [51].

Gibberellic acid was determined by triplicate in a Single Quadrupole LC/MS interfaced with an Agilent 1200 series liquid chromatography system with a diode-array (UV-Vis) detector. An Eclipse plus C18 column with a particle size of 5 µm was used. AG 1, 3, and 7 standards (Sigma-Aldrich, St. Louis, MI, USA) were used in HPLC analyses essentially as described by [12].

### 4.6. Effect of B.c-A, in Combination with SiO_2_-NPs on Chili Pepper

To evaluate the bacteria alone or in combination with SiO_2_-NPs in a living vegetal crop, chili pepper plants were established in a greenhouse with different treatments. Germination percentage, plant height, stem thickness, number of leaves, number of fruits, fruit yield, and CAT and PAL enzymatic activities were evaluated.

In the germination test, 15 seeds by triplicate were treated with 5 different applications: (a) deionized water (Water), (b) *B.c*-A, (c) SiO_2_-NPs, (d) *B.c*-A+ SiO_2_-NPs 10 ppm, and (e) *B.c*-A+ SiO_2_-NPs 100 ppm. Twenty milliliters of each treatment was prepared, and 50 seeds were immersed in each one. The seeds were soaked in each separated solution for 1 h, and after this, the seeds were placed on sterile Petri dishes with filter paper to maintain the humidity. Petri dishes were closed and incubated at room temperature (25 ± 3 °C) with a relative humidity of 17% for 5 days. Germination percentage was calculated by counting the daily number of germinated seeds by treatment [52].

Additionally, the treatments were evaluated in seedlings. Nine plants per treatment were established randomly in a 60 m^2^ greenhouse for 8 weeks. The treatments applied were (a) Water, (b) *B.c*-A, (c) SiO_2_-NPs, and (d) *B.c*-A+ SiO_2_-NPs 100 ppm. Seedlings were transplanted into plastic bags with inorganic substrate when 2 true leaves were visible. Drop-by-drop irrigation was applied automatically with standard Steiner nutrition.

Treatments were applied at the drop point with a plastic atomizer 1 week after the transplant and every 15 days. Growth, development, and yield variables were determined with the following variables: height of the plant, height, thickness of stem, number of leaves, flowers, and fruits per plant, and also the weight and size of fruit at first harvest 40 days after transplant. Finally, 2 leaves of 3 plants per treatment were collected at 5 min and 1 and 2 months after treatment application and stored at −80° C for further processing. The stored leaves were macerated in a phosphates buffer and centrifuged at 12,000 rpm, and the supernatant was used as EE for PAL and CAT determination.

### 4.7. Statistical Analysis

The obtained results from survival, phosphate solubilization capacity, enzymatic activities, and gibberellin production of the bacteria were analyzed by one-way variance analysis (ANOVA). Data from chili pepper seed germination percentage, leaves enzymatic activities, and physiological variables were analyzed by two-way variance analysis (ANOVA). Means were compared via Tukey’s test (*p* = 0.05). Statistical analysis was performed with JMP PRO 16 software( Version 16.2, JMP Statistical Discovery LLC, Cary, NC, U.S.).

## 5. Conclusions

Based on the results, it is concluded that the bacteria *B.c*-A was elicited with SiO_2_-NPs 100 ppm, increasing its phosphate solubilizing activity, gibberellin 7 (GA7) production, and SOD and CAT activities. In addition, the combined application of *B.c*-A and SiO_2_-NPs 100 ppm improved the germination percentage of chili pepper seeds and increased the number of leaves, plant height, number of fruits, total crop yield, and antioxidant activity of chili pepper. These beneficial properties may be attributed to the elicitation of the bacteria with nanoparticles since treatments performed separately did not show the same results. However, the PGPB–NP–plant interaction is complex, so different studies will be necessary to gather the information that allows the establishment of the physical, biochemical, and genetic mechanisms of such interaction. Moreover, this priming effect of SiO_2_-NPs on *B.c*-A might be an interesting strategy to be evaluated in future PGPB formulations for practical agricultural uses, enhancing the effect of PGPB treatments in agriculture and horticulture.

## Figures and Tables

**Figure 1 plants-11-03445-f001:**
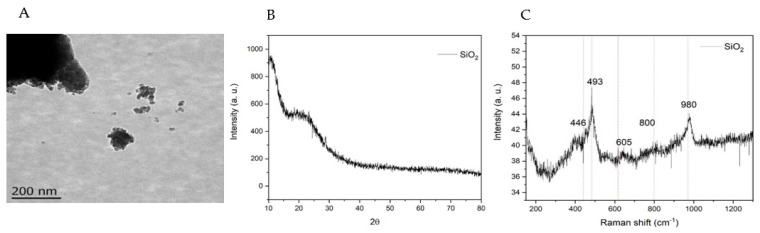
SiO_2_-NPs physicochemical characterization. (**A**) TEM image of SiO_2_-NPs, (**B**) XRD pattern, and (**C**) Raman spectra of the SiO_2_-NPs.

**Figure 2 plants-11-03445-f002:**
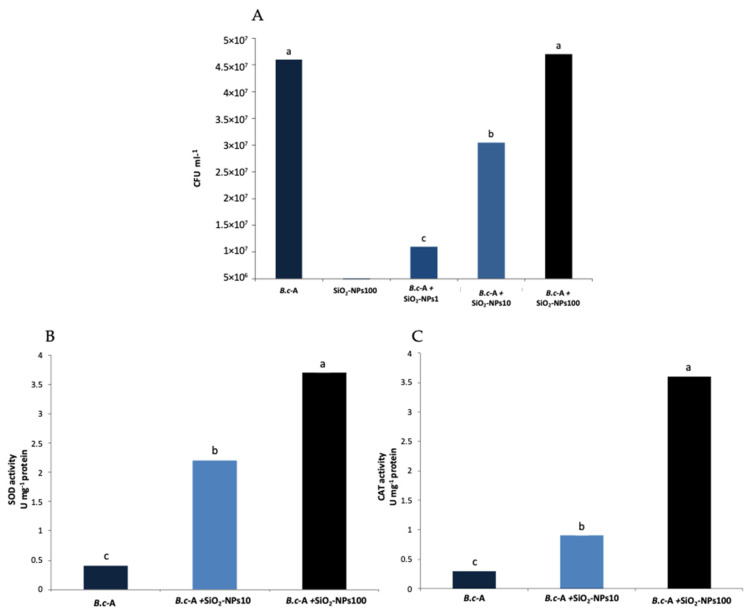
Effect of SiO_2_-NPs on *B.c*-A, after 24 h of exposure. (**A**) *B.c*-A survival. (**B**) SOD antioxidant activity of *B.c*-A. (**C**) CAT antioxidant activity of *B.c*-A. Different letters indicate significant differences according to Tukey’s test (*p* = 0.05).

**Figure 3 plants-11-03445-f003:**
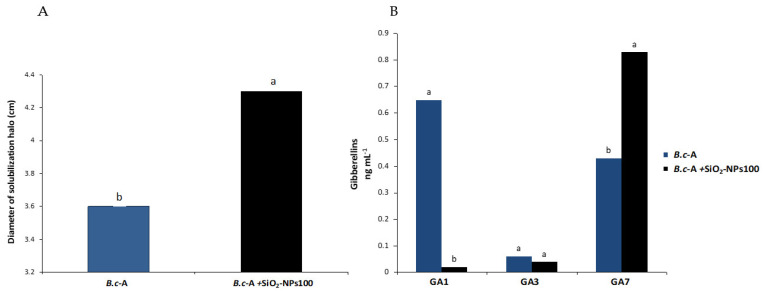
In vitro effect of SiO_2_-NPs on *B.c*-A plant-growth promoting activity after 24 h of exposure. (**A**) *B.c*-A phosphate solubilization activity using Pikovskaya medium culture. (**B**) *B.c*-A gibberellins production. Different letters indicate significant differences according to Tukey’s test (*p* = 0.05).

**Figure 4 plants-11-03445-f004:**
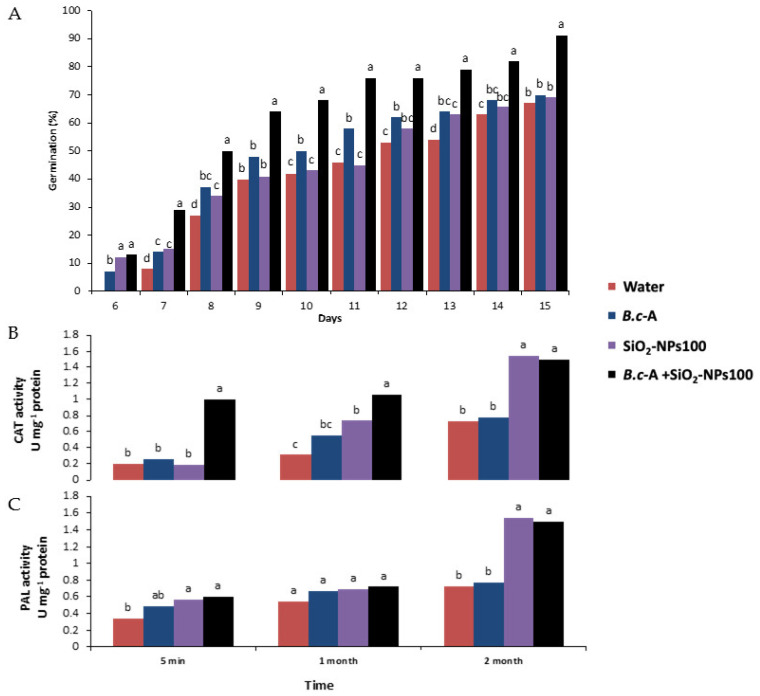
Effect of SiO_2_-NPs on germination and antioxidant activity of chili pepper. (**A**) Germination percentage of chili pepper seeds. (**B**) CAT activity of chili pepper plant; (**C**) PAL activity of chili pepper plant. Different letters indicate significant differences according to Tukey’s test (*p* = 0.05).

**Figure 5 plants-11-03445-f005:**
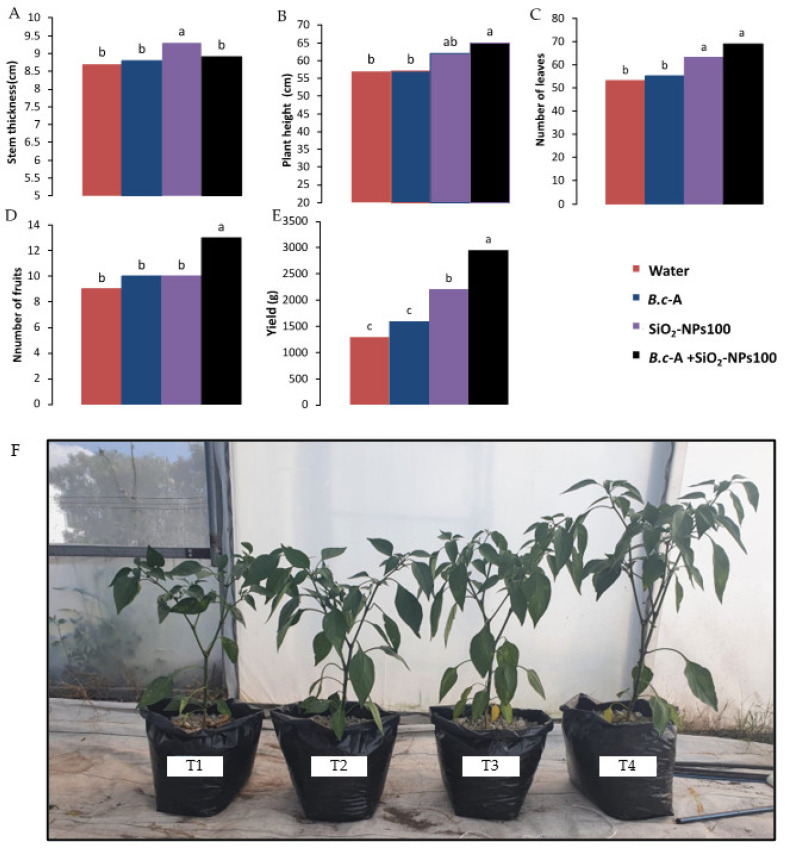
Physiological evaluation of the effect of SiO_2_-NPs on chili pepper after 8 weeks. (**A**) Stem thickness, (**B**) plant height, (**C**) number of leaves, (**D**) number of fruits, (**E**) yield of chili pepper, (**F**) phenotypical differences of height in chili pepper among the treatments. T1 = water, T2 = *B.c*-A, T3 = SiO_2_-NPs 100 ppm, T4 = *B.c*-A + SiO_2_-NPs 100 ppm. Different letters indicate significant differences according to Tukey’s test (*p* = 0.05).

## Data Availability

Not applicable.

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
