# Peer review of "Elicitation of *Bacillus cereus*-Amazcala (*B.c*-A) with SiO_2_ Nanoparticles Improves Its Role as a Plant Growth-Promoting Bacteria (PGPB) in Chili Pepper Plants"

_plants, 2022, doi:10.3390/plants11243445_

Round 1
Reviewer 1 Report
The manuscript describes the effect of bacteria-NPs as a PGPB in chili pepper. The authors investigated its effects on physiological and agricultural traits including CAT/PAL enzymatic activity, phosphate solubilization capacity and seed germination, in Pepper plants. SiO2-NP2 100 ppm treatment as a eustressor could induce defense related responses and improve chili pepper growth. It is beneficial to tolerance research and utilization in pepper plants. So I think it could be accepted with some minor revisions.
Suggesting:
1. Total 10 figures should be conbimed into 3-5 figures.
2. Please attention to the Reference format.
Author Response
Referee 1:
Comments and Suggestions for Authors
The manuscript describes the effect of bacteria-NPs as a PGPB in chili pepper. The authors investigated its effects on physiological and agricultural traits including CAT/PAL enzymatic activity, phosphate solubilization capacity and seed germination, in Pepper plants. SiO2-NP2 100 ppm treatment as a eustressor could induce defense related responses and improve chili pepper growth. It is beneficial to tolerance research and utilization in pepper plants. So I think it could be accepted with some minor revisions.
Suggesting:
Referee 1 comment: Total 10 figures should be combined into 3-5 figures.
Response: Dear Referee 1. We appreciate your comments. We have combined our figures to obtain 5 in total. Thanks for your comment.
Referee 1 comment: Please attention to the Reference format.
Response: We have noticed some errors in our reference format and corrected them. Thanks for your comment.

Reviewer 2 Report
In Introduction, please add more about the biochemical and physiological role of “SiO2-NPs” in the Introduction.
Why the treatments are different in different figures?
Fig 4.: Please write SOD activity and CAT activity in Y-axis.
Fig.6: Please write Gibberellins in Y-axis.
Fig. 5: “phosphate solubilizing capacity” of what? Where? Please mention/write. Please check the lettering of this Fig.
Figure 11. Please write CAT and PAL activity in Y-axis.
Line 229-244: Please discuss different data separately.
Line 245-253: This part can be replaced by the relevant research findings (GA production)
Line 254-258: The reason behind bacteria and SiO2-NPs -induced increase of number of leaves, plant height, number of fruits, and total crop yield was not discussed. Please discuss.
Why and how the enzyme activity changes should be discussed.
Line 289- 302: “NPs can improve the beneficial attributes of PGPBs”- Please discuss the beneficial attributes and the physiological/biochemical reasons.
Conclusion should be rewritten stating the results in brief, limitations of the present study and future perspectives.
Author Response
Referee 2:
Comments and Suggestions for Authors
Referee 2 comment: In Introduction, please add more about the biochemical and physiological role of “SiO2-NPs” in the Introduction.
Response: Dear Referee 2. We appreciate your comments. We have added to the introduction more about the biochemical and physiological role of SiO2-NPs. You can find it on lines 79-92.
Referee 2 comment: Why the treatments are different in different figures?
Response: In the in vitro experimental part, different concentrations of SiO2-NPs were applied to evaluate their effect on the bacteria. So these treatments were initially different. However, in the following tests, only the SiO2-NPs 100 ppm concentration was tested. In the case of the germination trial, an extra dose (10 ppm) was tested to gather more information. However, to avoid confusion, we have eliminated this treatment since it does not represent relevant information for the study.
Referee 2 comment: Fig 4.: Please write SOD activity and CAT activity in Y-axis.
Response: We have added SOD activity and CAT activity in Y-axis. Figure 4 is now Figure 2-B, C. Lines 156-158.
Referee 2 comment: Fig.6: Please write Gibberellins in Y-axis.
Response: We have added Gibberellins in Y-axis. Figure 6 is now Figure 3-B. Lines 170-173.
Referee 2 comment: Fig. 5: “phosphate solubilizing capacity” of what? Where? Please mention/write. Please check the lettering of this Fig.
Response: We have corrected this figure. Figure 5 is now Figure 3-A. Lines 170-173.
Referee 2 comment: Figure 11. Please write CAT and PAL activity in Y-axis.
Response: We have added CAT and PAL activity in Y-axis.Figure 11 is now Figure 4-B, C. Lines 192-195.
Referee 2 comment: Line 229-244: Please discuss different data separately.
Response: We have rewritten this part of the paper to discuss different data separately. Lines 236-274
Referee 2 comment: Line 245-253: This part can be replaced by the relevant research findings (GA production)
Response: We have added relevant information about GA production. Lines 279-284
Referee 2 comment: Line 254-258: The reason behind bacteria and SiO2-NPs -induced increase of number of leaves, plant height, number of fruits, and total crop yield was not discussed. Please discuss.
Response: We have discussed why bacteria and SiO2-NPs -induced beneficial effects on chili pepper plants. Lines 302-318.
Referee 2 comment: Why and how the enzyme activity changes should be discussed.
Response: We have discussed the enzyme activity changes in chili pepper plants. Lines 320-331
Referee 2 comment: Line 289- 302: “NPs can improve the beneficial attributes of PGPBs”- Please discuss the beneficial attributes and the physiological/biochemical reasons.
Response: We have discussed the beneficial attributes of PGPBs elicited by NPs, and some possible reasons. Lines 334-339. Moreover, this part was described throughout the discussion when we discussed the elicitor effect of NPs on the bacteria.
Referee 2 comment: Conclusion should be rewritten stating the results in brief, limitations of the present study and future perspectives.
Response: We have rewritten the conclusion. Lines 462-474

Round 2
Reviewer 2 Report
Language can be improved